# Anthropogenic Changes in a Mediterranean Coastal Wetland during the Last Century—The Case of Gialova Lagoon, Messinia, Greece

**Giorgos Maneas** [1,2,*]**, Eirini Makopoulou** [1]**, Dimitris Bousbouras** [3]**, Håkan Berg** [1] **and Stefano Manzoni** [1,4]

[1]  Department of Physical Geography, Stockholm University, 10691 Stockholm, Sweden; eirinimakop@gmail.com (E.M.); hakan.berg@natgeo.su.se (H.B.); stefano.manzoni@natgeo.su.se (S.M.)
[2]  Navarino Environmental Observatory, 24001 SW Messinia, Greece
[3]  Hellenic Ornithological Society, 10681 Athens, Greece; bousbourasd@gmail.com
[4]  Bolin Centre for Climate Research, 10691 Stockholm, Sweden
[*]  Correspondence: giorgos.maneas@natgeo.su.se; Tel.: +30-6979809570

**Abstract:** Human interventions during the last 70 years have altered the characteristics of the Gialova Lagoon, a coastal wetland that is part of a wider Natura 2000 site. In this study, we explore how human interventions and climate altered the wetland's hydrological conditions and habitats, leading to changing wetland functions over time. Our interpretations are based on a mixed methodological approach combining conceptual hydrologic models, analysis of aerial photographs, local knowledge, field observations, and GIS (Geographic Information System) analyses. The results show that the combined effects of human interventions and climate have led to increased salinity in the wetland over time. As a result, the fresh and brackish water marshes have gradually been turned into open water or replaced by halophytic vegetation with profound ecological implications. Furthermore, current human activities inside the Natura 2000 area and in the surrounding areas could further impact on the water quantity and quality in the wetland, and on its sensitive ecosystems. We suggest that a more holistic understanding of the broader socio-ecological system is needed to understand the dynamics of the wetland and to achieve sustainable long-term management and conservation strategies.

**Keywords:** wetland; human interventions; Mediterranean climate; hydrology; salinity; land use/land cover change; vegetation; agriculture; tourism; Natura 2000

## 1. Introduction

Greece has lost more than 70% of its wetlands over the last 130 years due to policymakers having decided to drain lakes and coastal lagoons to increase agriculture land areas [1,2]. Further, the introduction of new mechanized technologies, the increased use of chemical fertilizers and pesticides, and governmental policies that favour maximizing production in agriculture, have impacted on soil and water resources, diminishing the benefits that wetlands provide [3,4]. Furthermore, water resources are increasingly used and affected by human activities (e.g., irrigation, pollution from agrochemicals and industry by-products), resulting in lower availability and quality of fresh water sustaining wetland ecosystems [5].

Gialova Lagoon wetland (GLw), a coastal wetland located in Greece, is an example of a wetland area that, since the 1960s, has suffered from extensive drainage and the impact of other human activities [6–8]. These activities have resulted in severe environmental changes, leading to frequent dystrophic crises in the lagoon [9,10]. Several studies have focused on different physical and biological characteristics of the lagoon [1,9–16]. Yet, considering the lagoon is part of a larger wetland that

provides important aquatic habitats within a Natura 2000 site, previous studies have neglected the neighboring wetland area and its surroundings.

A Special Environmental Study (SES), finalized in 2000, highlighted the importance of the GLw. Funded by an EU (European Union) LIFE-nature project, and implemented through the Natura2000 framework, the SES provided guidelines for water uses, species, and habitat conservation and for human activities, such as fishing, agriculture and tourism [17]. In terms of species conservation, major attention was given to the protection of the African Chameleon (*Chamaeleo africanus*) since the area hosts the only European population. Prevention of illegal bird hunting, conservation actions and increasing awareness among the local inhabitants were also high on the agenda [7,17]. Infrastructure for educational purposes (e.g., paths, botanical and habitat signs, information centre) and for promoting eco-touristic activities (e.g., observation hides) were also constructed [17].

Despite its well-intentioned aims, as these guidelines were not institutionalized by the Greek state, the management goals remained on paper, and the lack of political will led to conflicts. However, the Hellenic Ornithological Society (HOS) continued to work in the area aiming to keep the socio-ecological system in a relative balance. Still, conflicts with the local community arose. The HOS project ended in 2012, and since then, the area has suffered from a lack of clear strategic management and concrete actions [7].

Conflicts related to policies, socioeconomic factors and conservation are thought to be the main factors hampering the implementation of the Natura 2000 framework in Greece [18]. These conflicts have been linked to the fact that stakeholder participation in the management of these sites had been limited [19]. In a similar way, the designation of GLw as a Natura 2000 site has followed a top-down administrative, expert-based and protectionist approach leading to conflicts.

A more integrative approach considering land use history and changes, and the practices and relationships of local communities with natural resources, could create the basis for a better understanding of the socio-ecological system [20]. In turn, this could pave the way for future research and participatory processes that ensure the sustainable management of GLw and other Natura 2000 sites.

With consideration of the above, it is important to identify and understand the causal links between human interventions and the dynamics of GLw. This step has not been undertaken, and motivates the aim of this research. To fill this gap, we investigated how human interventions have affected the hydrology, water quality, ecology and land cover of the whole wetland area and surroundings, starting from the post-World War II years till today. To complement previous work, we offer a more holistic perspective on the socio-ecological context in which these interventions occurred and suggest that this is a critical element for effective management of the natural resources in the area.

## 2. Materials and Methods

### 2.1. Study Area

The Gialova Lagoon wetland (GLw) is a coastal wetland located in south-west Messinia, Greece (latitude: 36°58′, longitude: 21°39′). It is classified as an important bird area (IBA) and a wildlife refuge. It is also part of a wider Natura 2000 site, which includes Sfaktiria Island and the coastal sand dunes from Voidokilia to the north of the Costa Navarino resort (Figure 1). In addition, it is also characterised as an archaeological protected site. The boundaries of the study area are defined at the north and east by two canals constructed during the 1960s. Towards the west, a rocky hill with historical importance (also known as Palaiokastro), and the semi-enclosed Voidokilia Bay, separate the area from the Ionian Sea. A 3.3 km long and approximately 0.15 km wide natural sand formation, also known as the Divari sand barrier, separates the area from the semi-enclosed Navarino Bay to the south. The area is comprised of a saltwater lagoon and wetland areas as well as cultivated lands (Figure 1). In this study, we will refer to this area as the GLw-Natura2000 area (green line).

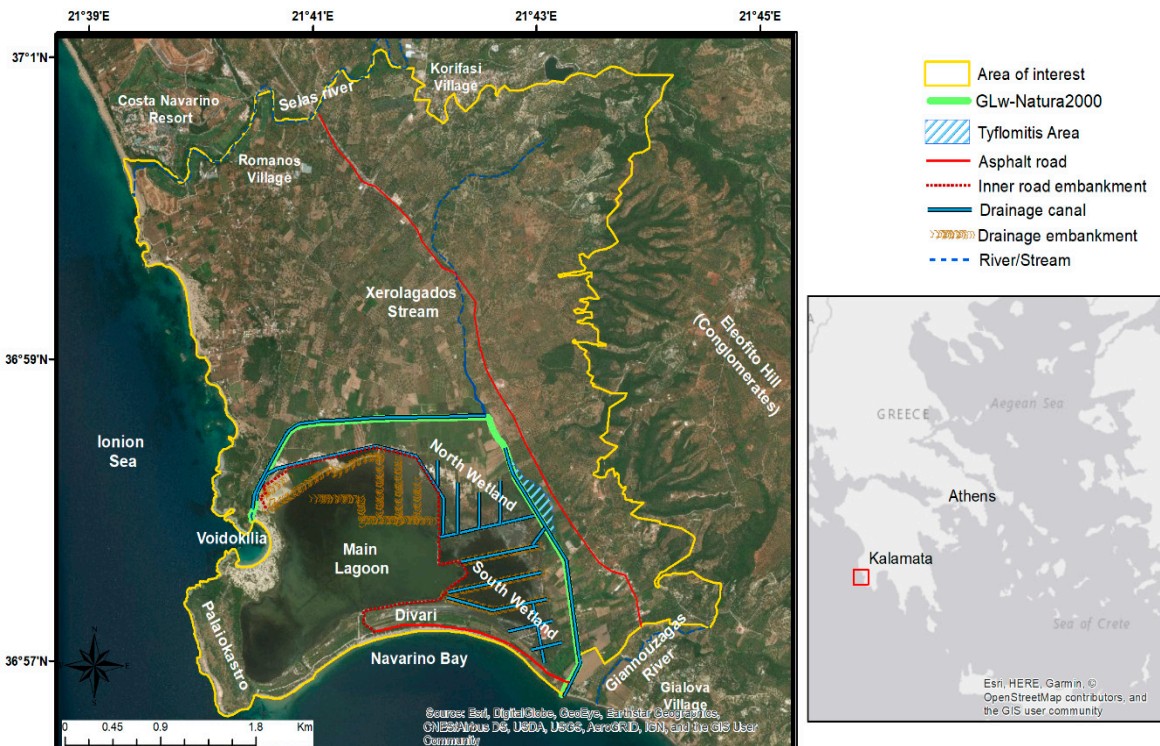

**Figure 1.** Map of the study area, including the Gialova Lagoon (GL) and surrounding wetlands in the Natura 2000 site (green line), and the wider area of interest encompassing the catchments draining into GL (yellow line).

The GLw is part of the Navarino embayment, which is a 10 km long and up to 4 km wide alluvial plain [21]. Under the conglomerates of the Eleofito hills at the east of the GLw, lies a groundwater aquifer [7,22]. The conglomerates continue all the way to the east side of the wetland under the alluvial sediments, but they have been submerged due to past tectonic activity. This sinking has resulted in a surface freshwater supply from up-welling groundwater due to the pressure differences forming a frontier known as the Tyflomitis artesian springs area (Figure 1 and Figure S3). The Tyflomitis area is the main freshwater provider of the wetland.

## 2.2. Assessing the Impacts of Human Interventions and Climatic Factors on Hydrology and Salinity

Our analysis covers a timespan of more than 70 years and is divided into three time periods. The periods are defined as:

- Reference period (pre-intervention period)—from 1945 to 1960, prior to the drainage effort and other major human interventions.
- Period of interventions—from 1961 to 1999, during which the wetland was isolated from any source of surface freshwater inputs, and other interventions also occurred.
- Present period—from 2000 to present, starting after the last human interventions, which resulted in a re-supply of surface fresh water to the wetland.

### 2.2.1. Aerial Photographs Interpretation

A set of aerial photographs, acquired by the Hellenic Military Geographical Services, were used as base maps to reconstruct the hydrological conditions over time, and conceptual schematics were used to interpret and present the prevailing hydrological conditions in the different time periods. The 1945 aerial photograph was used to establish reference conditions prior to any major interventions. Aerial photographs from 1972, 1980, 1992 and 1999, and a Google Maps image from 2010 [23], were

used to reconstruct the hydrological conditions prevailing after each human intervention in the Glw-Natura2000 area.

To fill knowledge gaps, a modification of the participatory geographic information system (PGIS) methodology [24] was used to gather information on past land and water uses. Interviews were held with five local elderlies (two over 65 and three over 70 years old living in Gialova, a village of 275 residents [25]), who have been living and working in the area for all of their adult lives. These interviewees were considered as key informants (KIs) for describing past conditions and transformations over time. In addition to being interviewed, they were also asked to draw on the 1945 aerial picture the basic land cover and water uses before the period of interventions. Despite the limited number of KIs, we consider them as representative of the (small) elderly population in the area as they cover a broad range of occupations and history of interactions within the GLw-Natura2000 area. These interviews and drawings were not used as the primary information about the area, but complemented our interpretation of black and white (b/w) aerial photographs and understanding of the available 1952 historical map (Figure S1). To ensure that the information gathered from these interviews was robust, we only kept answers that were consistent among the KIs. Their drawings were used as inputs in the analysis of land use and habitats changes described below.

### 2.2.2. Climatic Data

Climatic factors were analyzed based on a monthly climatic dataset from the meteorological station at Methoni (latitude: 36°50′, longitude: 21°42′, elevation: 52 m, distance from the study site: 15.6 km), covering the period 1956–2011. For each period, the monthly temperature time series was used to estimate the evaporation rate from the wetland using the Thornthwaite method [26,27], with the addition of day-length modifiers. This method is suitable for estimating lake and wetland evaporation, representing a good balance between simplicity (it only requires temperature time series) and accuracy [26,27]. The obtained evaporation rates were expressed in mm per month or per year (depending on the analysis), consistent with the precipitation data. To estimate the seasonality of climatic parameters, the average monthly precipitation and the average monthly temperature for the period 1956–2011 (56 years) were calculated. The overall trends in precipitation, evaporation rate and temperature were evaluated by applying a simple linear regression model on total annual precipitation and mean annual temperature values for the entire study period.

To compare reference conditions and present conditions, we used climatic data for the periods 1956–1960 and 2000–2011, respectively. To account for changes in wetland size between the periods, hydrologic fluxes expressed in mm/year (or m/year) were converted to $m^3$/year by multiplying water depths by the time-varying wetland areas. The results from these calculations were used as inputs in the analysis of the salinity variations described below.

### 2.2.3. Stream Discharge

Xerolagados and Tyflomitis flow rates, in 1983–1984, were estimated at $1.6 \times 10^6$ $m^3$/year and $2.7 \times 10^6$ $m^3$/year, respectively [7]. A later study reported a Tyflomitis discharge below $2 \times 10^6$ $m^3$/year, lower than earlier estimates, probably due to increased water use for irrigation after 1990 [22]. Here we use the most recent estimates [28], in which the Xerolagados discharge was estimated at $5 \times 10^6$ $m^3$/year and Tyflomitis artesian springs at $1.58 \times 10^6$ $m^3$/year before the increase of water extraction (past period), but only at $0.5 \times 10^6$ $m^3$/year under present conditions (2009), indicating a decrease of almost 70%.

### 2.2.4. Conceptual Hydrologic and Salt Balance Models

To assess how salinity has been changing in GLw over time, we propose a conceptual model based on water volume and salt mass balance equations valid at the annual timescale. For the lagoon water volume $V$, we can write a balance equation as:

$$dV/dt = P + Rf + Rs + Gf + Gs - E \tag{1}$$

where $P$ is precipitation, $E$ is the evaporation rate, $R$ represents surface water exchanges and $G$ represents groundwater exchanges (all fluxes are expressed as m$^3$/year). Letters $s$ and $f$ refer to saline and fresh water, respectively. Specifically, $Rs$ is the flux of water exchanged with Navarino bay via the sea-lagoon channel (positive when water is entering the lagoon, negative when water is leaving), $Rf$ is runoff from Xerolagados and Tyflomitis ($Rf > 0$), $Gs$ is groundwater flow between the lagoon and the Ionian Sea (Voidokilia and Navarino Bays), which may be positive or negative depending on the direction of the water, and $Gf$ is the groundwater flow from the freshwater aquifers (assumed $Gf > 0$). Fluctuations in water level and open water area that define $V$ are affected by wind patterns and tides, but at the annual time scale they are negligible, allowing us to assume $dV/dt \approx 0$. As a result, we can re-write Equation (1) to highlight the dependence of saline water fluxes on freshwater fluxes as follows:

$$Rs + Gs = E - P - Rf - Gf \tag{2}$$

Equation (2) illustrates that increased freshwater inputs (from rainfall or streams) causes the saline water exchanges to be negative, that is, they promote loss of water from the lagoon to the sea.

The balance equation for the mass of salt $M$ reads:

$$dM/dt \approx V \times dCi/dt = CGs \times Gs + CRs \times Rs \tag{3}$$

where $CGs$ and $CRs$ are the salt concentrations associated with the fluxes $Gs$ and $Rs$. The first equality in Equation (3) is explained as follows: the mass of salt can be expressed as the product of salt concentration in the lagoon water $Ci$ and water volume, but because $dV/dt \approx 0$, we can approximate $d(V \times Ci)/dt \approx V \times dCi/dt$. It is important to note that both $Gs$ and $Rs$ can have different signs depending on whether water enters or leaves the lagoon. When water enters the lagoon from the Ionian Sea (Voidokilia and Navarino Bays) via groundwater or surface water, $CGs \approx CRs$, and both quantities are equal to the seawater salinity. For simplicity, we assume that the sea salinity fluctuations are negligible, because salinity varied only from 38.35 ppt to 38.65 ppt during the period 1946–2002 [29]. Equation (3) offers a framework for assessing how changes in hydrologic flows constrain changes in salinity in the lagoon. Any change that promotes outflows of saline water ($Gs < 0$ and $Rs < 0$), such as high rainfall and freshwater inputs (see Equation (2)), decreases $Ci$. In contrast, changes that promote inflows ($Gs > 0$ and $Rs > 0$), such as diversion of streams, increase $Ci$.

*2.3. Assessing the Impacts of Human Interventions on Land Cover and Land Uses*

To compare land cover and land uses between reference and present conditions, we constructed maps based on spatial data interpretations and field observations. The maps were built from digitized high-resolution satellite images available on the Google Earth platform and from the aerial photograph from 1945.

In order to reconstruct the characteristics of the area before the major human interventions and make it comparable to present, the 1945 aerial picture was georeferenced to the WGRS87/Greek Grid and used as a base map in ArcGIS (Version 10.5.Redlands, CA: Environmental Systems Research Institute, Inc., 2016). The information acquired by the KIs, as it was captured on their drawings (Figure S2) and in their statements, was then added according to the methodology described in Section 2.2.1. The information from the KIs was used to complement our understanding of reference conditions, and specifically about types of crops (inside and outside GLw-Natura2000 area) and vegetation (inside the GLw-Natura2000 area). The outcome from this analysis was the creation of a map showing the characteristics of the whole area of interest before the major interventions in 1960.

Field visits were conducted on a regular basis during the period July–August 2011 and in August 2018 to identify the major habitats and land uses inside the two canals draining the perimeter of the GLw-Natura2000 area. During field work, a portable GPS was used to mark the location of each observation. The observations were imported in ArcGIS for further analysis.

## 3. Results

### 3.1. Effects of Climatic Factors and Human Interventions on Hydrology and Salinity over Time

The hydrologic conditions of the wetland varied depending on:

- Water exchange between the wetland and the sea via a canal.
- Water exchange between the wetland and the atmosphere (evaporation and precipitation).
- Groundwater exchanges and inputs from inland aquifer.
- Freshwater inputs from nearby water bodies (Xerolagados stream and Tyflomitis spring area).

Because all the surface water exchanges have been heavily altered by human activities and because climatic conditions vary through time, we expect that the salinity of the lagoon and wetland has also varied in response to both anthropogenic factors and climatic variables, as described in the following sections.

#### 3.1.1. Human Alterations Affecting Hydrological Conditions over Time

Reference (Pre-Interventions) Period (1945–1960)

Based on our interpretation of aerial photographs, Figure 2 shows the surface water inputs to, and exchanges with, the wetland before the drainage effort. At that time, the Tyflomitis area was part of the wetland (main lagoon-wetland), the Xerolagados stream was flowing into the north east side of the wetland and there was at least one sea/lagoon channel allowing water exchanges with the sea. The KIs indicated that the sea/lagoon channel remained closed during the winter and was opened in the spring to let fish enter the lagoon and allow water exchange. Water from Tyflomitis springs flowed into the wetland throughout the year. The Xerolagados stream also supplied the wetland with fresh water during the winter season and until May–June (or later in some seasons).

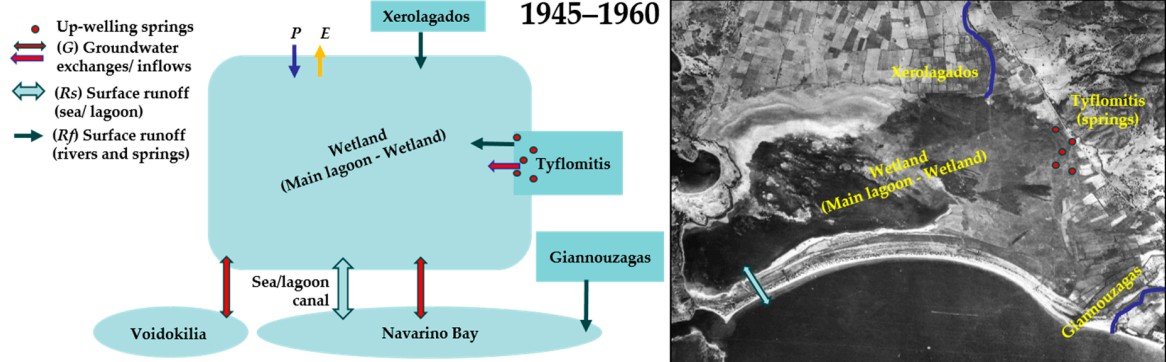

**Figure 2.** Schematic representation of the hydrologic conditions before the drainage interventions (reference period), based on a 1945 b/w aerial photograph and a map of the same period (Figure S1), complemented with information from KIs and available literature. Potential exchanges via the alluvial deposits and sand barriers are also presented. *P* and *E* are abbreviations for precipitation and evaporation, respectively.

Period of Interventions (1961–1999)

In 1960–1961, after a series of interventions, the hydrology of the wetland was altered. Water from the Tyflomitis area and the Xerolagados stream was diverted to flow directly into Navarino Bay and Voidokilia Bay respectively. A pumping station was built to lower the water level of the lagoon (purple arrow in Figure 3a), and deep channels and embankments were also constructed to drain the wetland (Figure 3a,b). As a result, in the summer of 1964, the former wetland area was completely dry. One of the KIs stated that "We could drive in the wetland," and added that after the abandonment of the drainage efforts (1964–1965), the wetland remained isolated from the sea and it became the source of bad odor and diseases like malaria. By 1964, the shape of the wetland had been changed to a deeper west side (main lagoon) and a shallower east side (wetlands) (Figure 3a).

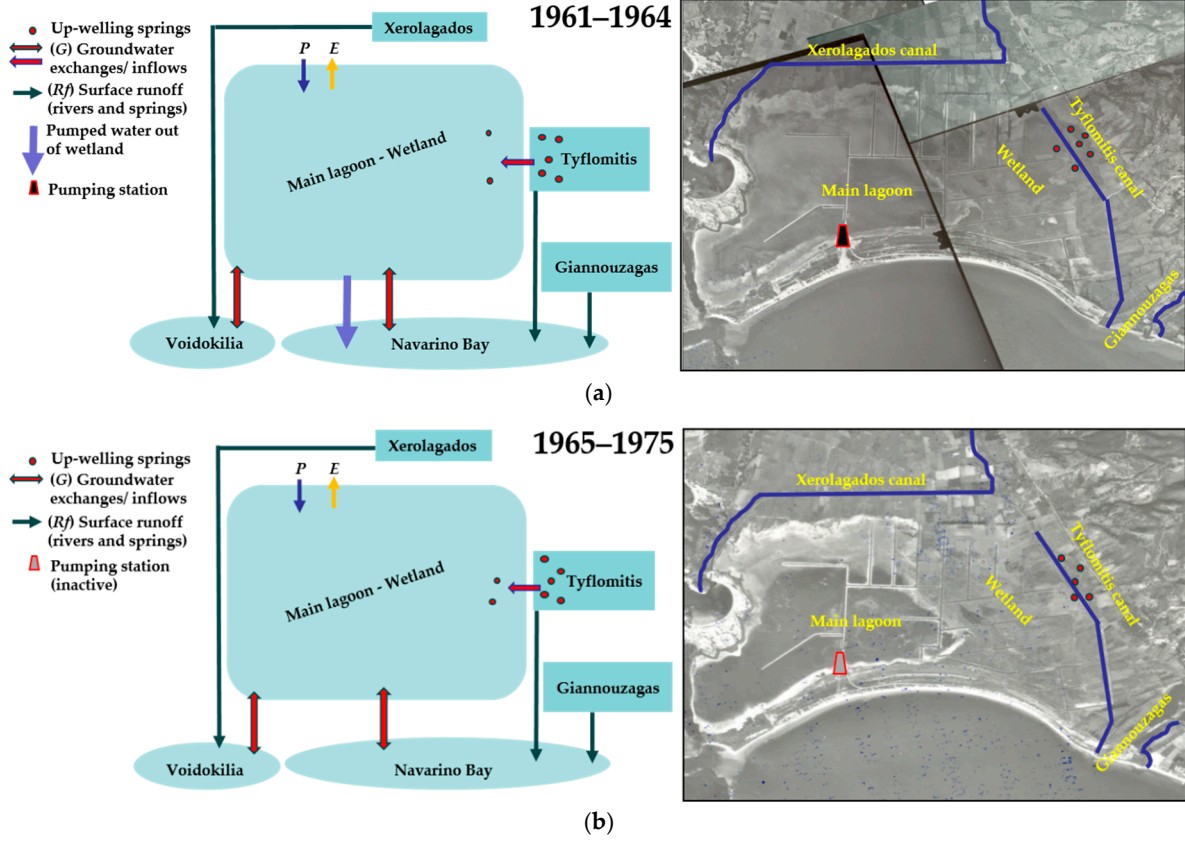

**Figure 3.** Schematic representation of the hydrologic conditions: (**a**) during the drainage period, based on a b/w aerial photograph from 1964, and (**b**) during the period 1965–1975, based on a b/w aerial photograph from 1972.

In 1976, the sea/lagoon canal was re-opened, and in 1979, it was deepened and widened by mechanical means to its present size. This allowed water to flow freely between the sea and the lagoon (Figure 4a). Also in 1979, construction of the inner embankment construction began [8], and by 1984–1985, it was extended all the way from the Voidokilia Bay to the pumping station (Figure 4b). This embankment subdivided the wetland into a main lagoon area and neighbouring wetlands, and prevented the mixing of these two water bodies [8]. In 1998–1999, two culverts were constructed in the inner embankment to allow freshwater inputs to the main lagoon [9,10,15].

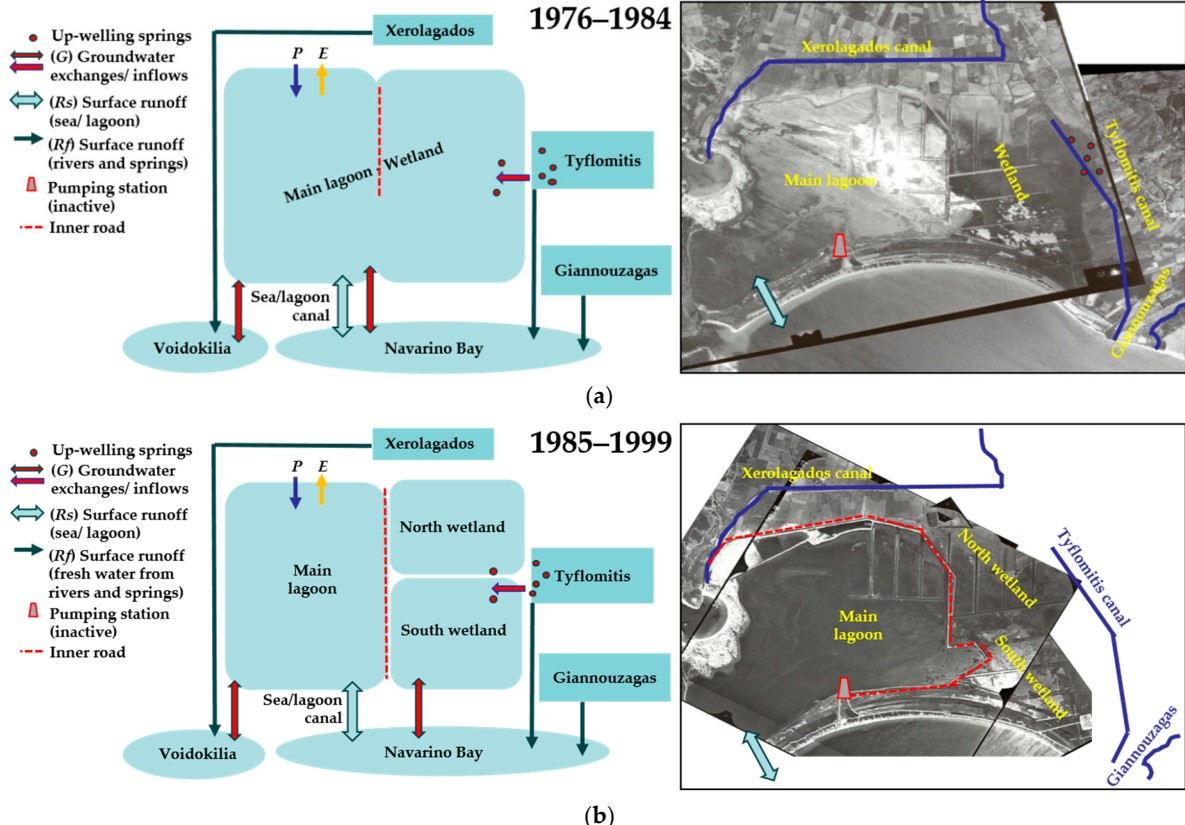

**Figure 4.** Schematic representation of the hydrologic conditions in the GLw-Natura2000: (**a**) during the 1976–1984 period based on b/w aerial photographs from 1980, and (**b**) during the 1985–1999 period based on b/w aerial photographs from 1992.

Present Period (2000–2018)

The present period starts after 1999–2000, when two sluice gates with a 1 m$^2$ cross section were constructed in the dykes of the two drainage canals to allow more fresh water into the wetland [17]. One gate was constructed to connect the Tyflomitis canal to the main lagoon (via a deep drainage canal), and one was constructed close to the end of the Xerolagados canal, on the north-west side of the main lagoon (via a drainage canal) (Figure 5a).

However, field observations showed that the Xerolagados sluice gate has remained blocked since 2010, interrupting the freshwater inputs from that canal (Figure 5b). Nonetheless, another culvert was constructed on the east side, connecting the Tyflomitis canal with the south side of the wetlands (Figure 5b). It was also evident that the drainage canal connecting the Tyflomitis canal with the main lagoon divided the wetland area in two, with a north and south part. There is one open inner culvert in each area allowing water exchange with the main lagoon. A third inner culvert, connecting the south side of the wetland to the main lagoon, was opened at the same time, but it was later blocked by accumulated sediments. Compared to past conditions, the water exchange with the sea, via the sea/lagoon canal, is now uninterrupted all year around.

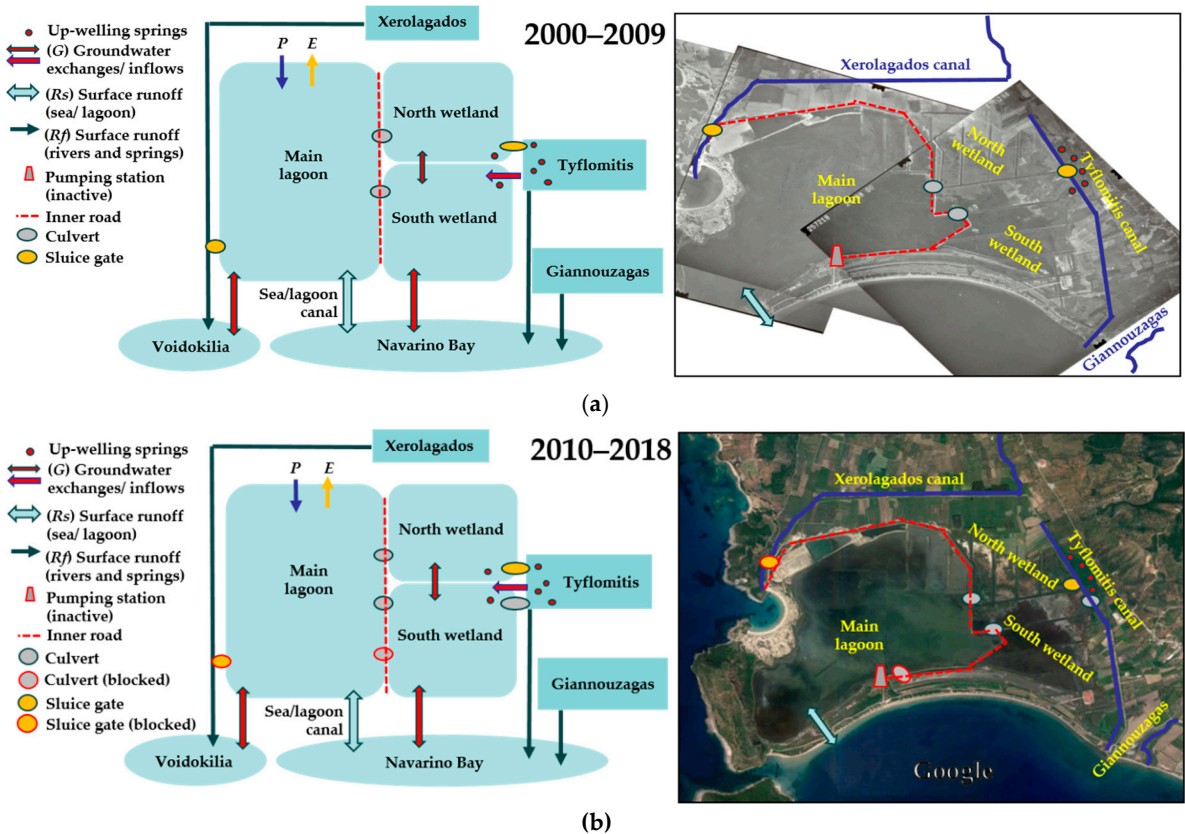

**Figure 5.** Schematic representation of the hydrological conditions in the GLw-Natura2000: (**a**) during the period 2000–2009 period based on a b/w aerial photograph from 2000, and (**b**) from 2010 to today based on a Google Earth image.

### 3.1.2. Climatic Variables over Time

Over the period 1956–2011, annual precipitation, evaporation rate and temperature exhibited mild decreases (based on linear regression), but none of these trends were statistically significant (the 95% confidence intervals for the slopes bracketed zero). The meteorological data showed clear seasonal trends for the temperature and precipitation, typical of a Mediterranean climate (Figure 6). The estimated evaporation rate followed the temperature trend, exhibiting a maximum during the summer months when precipitation is at its minimum. As a result, a seasonal water deficit developed during the summer and into the autumn, leading in most years to a net water deficit (on average approximately 200 mm/year).

### 3.1.3. Combined Effects of Human Interventions and Climate on Wetland Salinity

Based on the above analysis, we can compare the hydrologic conditions before the interventions to current conditions, and infer how salinity (and thus wetland ecology) has been impacted by compounded human and climatic changes. To this end, first the water balance of Equation (2) is expressed as a function of hydrologic fluxes in the two periods. Second, the differences between evaporation and precipitation (water deficits) are calculated for each period. Third, the water deficits and the water balances are used to estimate saline water inputs that drive changes in salinity.

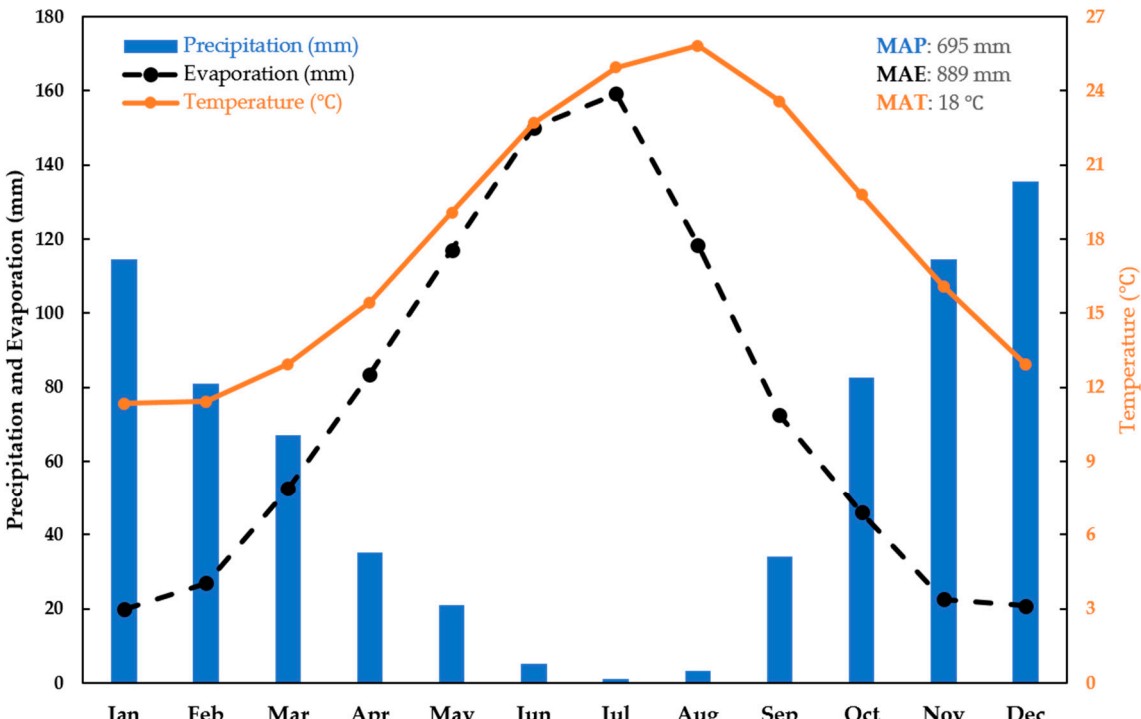

**Figure 6.** Monthly mean precipitation (MAP), evaporation rate (MAE) and temperature (MAT) for the period 1956–2011. Mean annual values for the whole study period are also reported in the top-right of the figure.

Before the 1960 interventions (reference-period), the salinity in the whole wetland was affected by surface freshwater inputs (*Rfpast*) from the Xerolagados stream and Tyflomitis springs, precipitation (*Ppast*), seawater exchanges with Navarino bay (*Rspast*), evaporation (*Epast*), groundwater inflows from Tyflomitis aquifer (*Gfpast*) and groundwater fluxes from Navarino and Voidokilia Bays (*Gspast*). Thus, during the reference period Equation (2) can be written as:

$$Rspast + Gspast = Epast - Ppast - Rfpast - Gfpast. \tag{4}$$

Because *Epast – Ppast* > 0, sources of water other than rainfall had to compensate for the water deficit in the lagoon. It is likely that during the wet season, the freshwater inputs (*Rfpast*, *Gfpast*) were prevalent, while in the dry season—due to the low stream and aquifer flows—saline water exchanges were more important. The quantification of these fluxes is not feasible with the available data.

At present time (2000–2018), the salinity in the wetland is affected by surface freshwater inputs (*Rfpre*) only from Tyflomitis canal (from the sluice door and the culvert), precipitation (*Ppre*), water fluxes through the sea/lagoon channel (*Rspre*), evaporation (*Epre*), groundwater inflows from Tyflomitis aquifer (*Gfpre*) and groundwater exchanges with the Ionian Sea (*Gspre*). Thus, for the present conditions Equation (2) can be written as:

$$Rspre + Gspre = Epre - Ppre - Rfpre - Gfpre \tag{5}$$

The water balances in Equations (4) and (5) were used to evaluate changes in salinity as driven by the balance of saline and freshwater fluxes exchanged by the lagoon. As explained in Section 2.2.3, fresh water provided by Xerolagados stream and Tyflomitis springs in the reference period amounted in total to *Rfpast* = 6.58 × 10⁶ m³/year. At present, even though a small fraction of the Tyflomitis discharge entered the lagoon (the rest flows to the Navarino Bay), for simplicity we assumed that *Rfpre* = 0.5 × 10⁶ m³/year. It was reasonable to assume that the groundwater inflows from the Tyflomitis aquifer (*Gfpre*) at present were also reduced at least by 70% and therefore *Gfpre* = 0.3 × *Gfpast*.

The Xerolagados sluice gate has been blocked since 2010, and before it remained closed during the most part of the year due to low water quality (polluted with by-products from the production of olive oil) [17]; therefore, we assumed for simplicity that *RfXerolagados* = 0 for the whole present period.

Figure 7 shows how climatic conditions varied during the period of human transformations. Overall, the annual evaporation rates were stable around 900 mm/year (presented as negative values in Figure 7 to indicate losses from the lagoon system). Precipitation decreased from the mid-1970s. However, inter-annual variability in each period was larger than the variability among periods. As a result, all periods were characterized by a net water deficit, but with strong inter-annual fluctuations, and some years with a small water excess. Since the climatic conditions have been relatively stable, the human interventions described above can be regarded as the main drivers of hydrologic change in the GLw area.

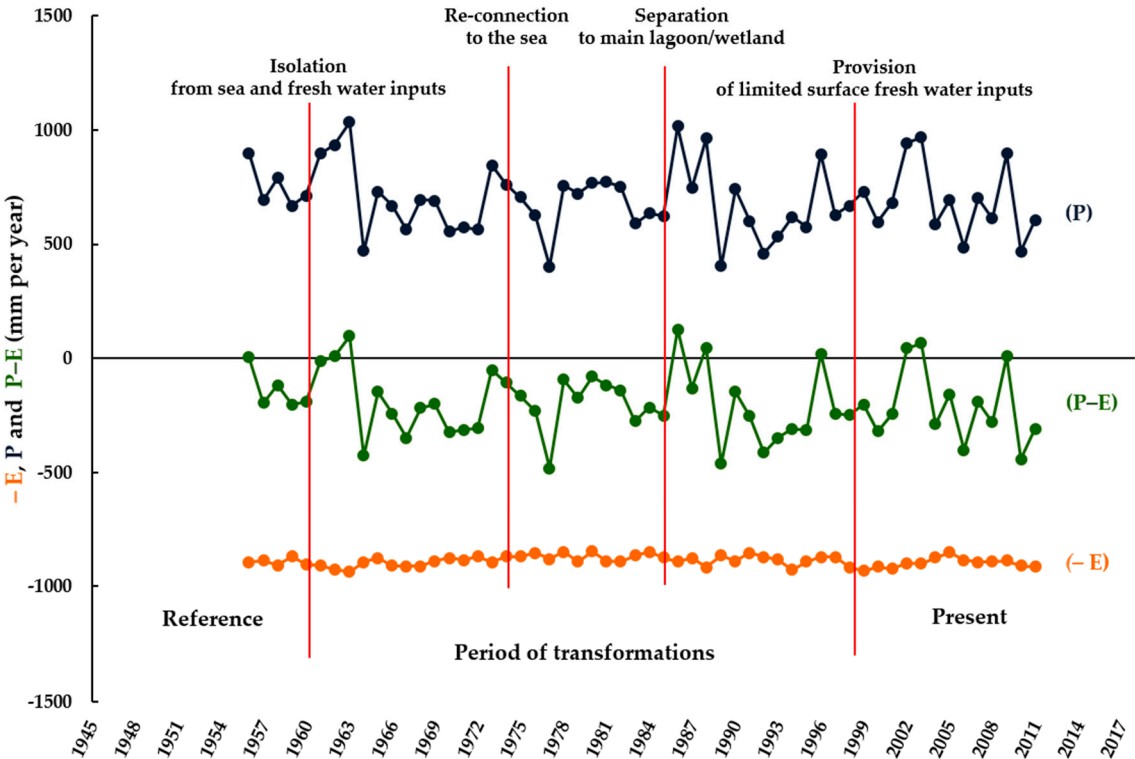

**Figure 7.** Annual water balance (mm/year) during the study period and major human interventions affecting surface water exchanges.

For reference and present periods, based on the available climatic data, the mean annual difference between evaporation and precipitation rates are estimated as:

$$Epast - Ppast = (0.893 - 0.752) \text{ m/year} \times 4.52 \times 10^6 \text{ m}^2 \approx 0.63 \times 10^6 \text{ m}^3/\text{year} \tag{6}$$

$$Epre - Ppre = (0.896 - 0.687) \text{ m/year} \times 4.18 \times 10^6 \text{ m}^2 \approx 0.87 \times 10^6 \text{ m}^3/\text{year} \tag{7}$$

Based on these estimates of water deficits and literature data presented above, we can re-write Equations (4) and (5) as follows:

$$Rspast + Gspast = -5.95 \times 10^6 - Gfpast < 0 \tag{8}$$

$$Rspre + Gspre = 0.37 \times 10^6 - 0.3 \times Gfpast > Rspast + Gspast \tag{9}$$

By comparing Equations (8) and (9), and based on the balance equation for the mass of salt (Equation (3)), it is reasonable to suggest that pre-intervention hydrologic conditions, compared to present, promoted outflows of saline water, decreasing the salinity and total salt mass in the wetland. Moreover, the opening of the sea/lagoon canal in 1976 and the absence of any surface freshwater inputs until 1998–1999, probably led to increasingly saline conditions in the wetland over the years. In fact, if we assume that water level fluctuations were negligible at the annual time scale ($dV/dt \approx 0$), the salinity must have been gradually increasing over time due to the annual freshwater deficit of about 200 mm. Without the natural freshwater sources, this deficit could only be met by seawater inputs. Indeed, in 1995–1996 salinity was on average 42.6 ppt, fluctuating between 26 ppt in the winter and 55 ppt in the summer [10,15]. These values are higher than those tolerated by the marshland vegetation that covered the area before 1960 (Section 3.2), lending support to the result of our conceptual model. The opening of the two sluice gates in 1998–1999 likely reduced the rate of the salinity increase. However, these freshwater inputs were probably not enough to reverse the situation and to significantly reduce the salinity.

### 3.2. Impacts of Human Interventions on Land Cover/Land Uses

During the study period, the characteristics of the area have been substantially transformed (Figure 8). When the KIs were asked to give an overall impression of the wetland status during the reference period (prior to 1960), they described it as an area covered with tall green vegetation and rich fauna (especially birds), with no barriers to water exchanges with inland surface water bodies. On the east side of the wetland, the KIs drew on the aerial photograph an area covered with cattail, reeds and tamarisks and described it as freshwater marshes. They also drew farms inside and near the wetland (today not included in the Natura 2000 area), in which farmers cultivated rice (Figure 8). Interpretation of the b/w aerial photographs allowed for the identification of mixed saline and freshwater ecosystems in the reference period. This marshland ecosystem can be regarded as the "natural" state prior to the major post-World War II interventions.

Fishing and bird-hunting were highlighted as major activities in the wetland and all the KIs mentioned that the area did not suffer from hunger during the Second World War, mainly due to fishing and hunting in the wetland. The area attracted hunters from different villages, and there were thousands of ducks, geese, swans and other water birds hiding in the reeds and cattail growing on the east section of the wetland. Fishing rights were assigned to local cooperatives of fishermen, but individuals were fishing in the area. Farms that were located near the wetland, and are included in the Natura 2000 area, were mainly used for cereal cultivation. The KIs mentioned that the whole area surrounding the wetland was mainly covered with cereal crops (Figure 8). In the months of May–June, some farmers used to cultivate rice, which depended on stable freshwater conditions. In the summer, families from nearby villages would visit the area to harvest cattail, a typical freshwater species, to be used as a raw material for the construction of chairs and other products. The area was also used for grazing by 200 to 300 cows throughout the entire year and by 1500 sheep, which were moved from nearby areas for wintering.

The wetland coverage in the reference period was estimated at 452 hectares and comprised the main lagoon, the surrounding marsh area and the Tyflomitis area (Table 1 and Table S1). However, the construction of the two perimeter canals (1960s) excluded surface freshwater inputs, and also interrupted the original continuum of the wetland area, isolating the Tyflomitis area from the rest of the wetland (Figure 3). The construction of the inner road-embankment in 1984–1985 further subdivided the wetland area into the main lagoon and the north and south wetlands, which remained isolated until 1998–1999 when sluice doors and culverts were opened (Figures 4 and 5).

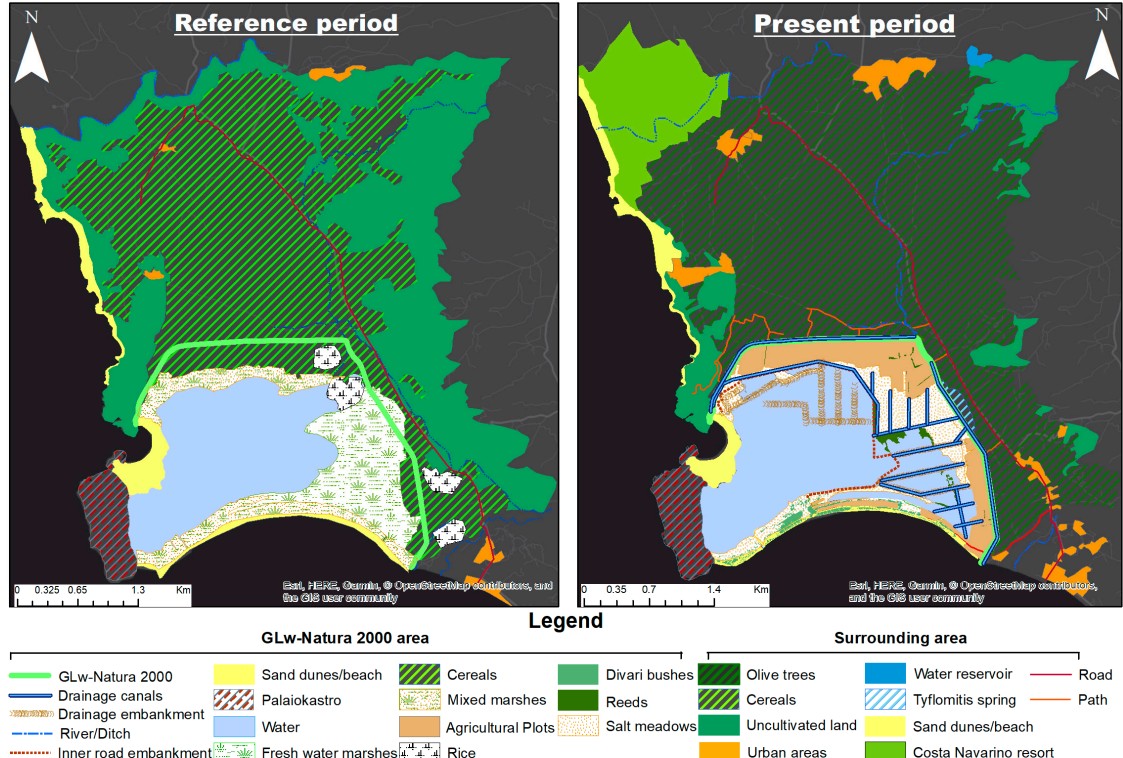

**Figure 8.** Land cover and major land uses in GLw-Natura2000 and the surrounding area in the past and present periods. The map of reference conditions was created in ArcGIS based on a 1945 b/w aerial photograph, interpretation of a historical map and interviews with the KIs. The map of present conditions was based on a 2010 Google Earth image and field observations in 2010 and 2018. The classification of the coverage of cereals (reference) and olive trees (present) in the surrounding area was based on the type of crop dominating the landscape.

From field observations, we identified that the marshes with their fresh or brackish water vegetation have been largely replaced by halophytic vegetation like *Salicornia europaea* and *Juncus spp.*, and some reeds remained at the sides of the canals (Figure 8). In the period 1976 to 1999, only sea water was entering the wetland and the increased salinity in the soil and water most likely drove this dramatic change in the vegetation. The KIs mentioned that the tamarisks in these areas were "burned by the salt" when sea water entered the wetland after 1976. In the 1980 aerial picture, part of the vegetation in the wetland area had already been replaced by open water (Figure 4a) and the process was completed by 1992 (Figure 4b).

At present, the size of the whole wetland has been reduced to 418 hectares, but its shape has changed. The continuum from lagoon to marshland, and to the wet areas of Tyflomitis have become a set of separate ecosystems with limited water exchanges. The northern wetland is mainly composed of relative deep drainage channels (with a maximum depth of 1.4 m), within which some reeds remain today, and salt meadows, which neighbor cultivations in the north. In the southern wetland, the coverage of reeds is also limited. This area is characterized by shallow waters (a maximum depth of 0.6 m during the winter), salt meadows, and old embankments (creating small parallel islands) covered by halophytes. The salt meadows tend to become partly dry during the summer and partly flooded during the winter. On one hand, open water coverage in the whole wetland has increased by 22.7% from 251 hectares in the past (lagoon) to at least 310 hectares at present (including the main lagoon and the shallow waters of the southern wetland, but not the water area of the northern wetland canals). On the other hand, the area covered with vegetation (marshes or/and salt meadows) has decreased by 46.9% from almost 194 hectares to only 103 hectares at present (Table S1). This combined evidence suggests that marshland vegetation has been substituted by open water (Figure 8).

**Table 1.** Area coverage (% of total studied area) of different classes in reference (prior to 1960) and present (2010) periods. The studied area includes the GLw-Natura2000 and the Tyflomitis area. The percentage coverage of each class before and after human interventions is given in the right column and last line, respectively. Areas that do not change are displayed on the diagonal of the table. Changes (gains or losses) in area coverage are displayed in the cells outside the diagonal (e.g., 13.3% of the area belonged to the W-M class in the reference period, but became part of the W-OW class in the present period). Errors attributable to uncertainties in the definition of the land use polygons amount to less than 0.5% of the total area.

| Period | | | Present Period | | | | | | |
|---|---|---|---|---|---|---|---|---|---|
| | GLw-Natura2000 Area | | W-TA | W-OW | W-M [1] | SD | RA | AGR | TOT |
| Reference period | Wetland—Tyflomitis Area | W-TA | 1.2 | | | | | 0.1 | 1.3 |
| | Wetland—Open Water | W-OW | | 36.5 | 2.9 | | | 1.2 | 40.6 |
| | Wetland—Marshes | W-M [1] | | 13.3 | 13.8 | | | 4.2 | 31.3 |
| | Sand Dunes | SD | | | | 5.9 | | | 6.0 |
| | Rocky Areas | RA | | | | | 8.4 | | 8.4 |
| | Agriculture | AGR | | | | | | 12.4 | 12.4 |
| | Total area | TOT | 1.2 | 49.8 | 16.7 | 5.9 | 8.4 | 17.9 | 99.9 [2] |

[1] W-M in the present period refers to salt meadows and some reeds which remain at the sides of the canals, while in the past period refers to fresh and brackish water vegetation. [2] When compared to the past, 0.8 hectares were lost from the Divari sand barrier due to erosion (west side).

Agricultural land inside the GLw-Natura2000 area has increased by 44.5% from 1960 to 2010 (Table 1). New cultivations were identified on the north-east and south-east areas of the wetland and on the south sand barrier (divari). compared to past conditions, when wheat farming was dominant and rice was also cultivated, 23% of the agricultural area is now cultivated with olive trees and 36% with horticultural or grain crops. other activities in the area include fruit trees cultivation (1.7%) and grazing (6.7%) by a smaller number of animals (4–5 cows and around 100 sheep) than in the past. A total of 7.4% is either fallow or uncultivated land.

In the surroundings, cereal crops have been replaced over time by irrigated and non-irrigated olive trees, which dominate the landscape and have been expanded even to areas that were originally characterized as uncultivated land (Table 2). This change was witnessed by the KIs: "Apart from the newly planted olive trees, many wild olive trees were grafted and transformed to commercial varieties creating a forest of olive trees in the area". The same trend can be seen in the whole Messinia region, which is characterized by extensive monoculture of olive trees (Figure S4).

**Table 2.** Area coverage (percentage with respect to the total wider area) of different classes in the wider area in reference (prior to 1960) and present (2010) periods. The percentage coverage of each class before and after human interventions is given in the right column and last line respectively. Surfaces that do not change (persistence) are displayed on the diagonal of the table. Changes (gains or losses) in area coverage are displayed in the cells outside the diagonal (e.g., 25.5% of the area belonged to the UCL class in the reference period, but became part of the CL class in the present period). Errors attributable to uncertainties in the definition of the land use polygons amount to less than 0.5% of the total area.

| Period | | | Present Period | | | |
|---|---|---|---|---|---|---|
| | Wider Area | | CL | UCL | UA [2] | TOT |
| Reference period | Cultivated Land | CL | 56.3 | | | 56.3 |
| | Uncultivated Land [1] | UCL | 25.5 | 13.7 | 3.0 | 42.3 |
| | Urban Areas | UA | | | 1.4 | 1.4 |
| | Total area | TOT | 81.8 | 13.7 | 4.4 | 100.0 |

[1] Estimated uncultivated land in both periods includes the areas which are neither cultivated nor urban, as well as the sand dunes found at the west coast of the wider area of interest, which is marked with yellow line in Figure 1. [2] UA at present also includes the part of Costa Navarino resort, which is found south of the Selas river, and is within our study area.

Another major shift in land uses, which was brought up by the KIs, was tourism. This shift started in the late 1970s. The number of tourists in Messinia has increased by almost 125% from approximately 120,000 arrivals in 1980 to nearly 280,000 in 2017 (Figure S5). The authors have witnessed tourism expansion in the area since 2010 (e.g., hotels and associated enterprises). This trend has led to a recent increase in urban areas (Table 2 and Table S2). Furthermore, within the GLw-Natura2000 area, in 2010, only one canteen operated in the area, while in the summer of 2018, there were three canteens and one watersport facility operating on the Divari sand barrier. Vehicles parked on top of shrubs and on the sand dunes of Divari and Voidokilia beach, and motocross racing along the dunes, were also recorded.

## 4. Discussion

Agricultural development, as well as general human appropriation of water resources, have led to global-scale changes in the water cycle [30] and wetland functions and distributions [31–34]. In Greece, expansion of irrigation has significantly lowered runoff [35], resulting in less water being available for wetlands downstream of irrigated areas. In the context of increased water abstraction, the Glw-Natura2000 area represents an example of complex and profound anthropogenic changes during the last century. While human activities have probably affected soils and hydrology in the area at least since the Mycenaean era [36], post-World War II alterations have had unprecedented consequences. These changes have affected both hydrologic fluxes and other characteristics of the area, but their implications on wetland functions and their feedbacks on the surrounding area have not been addressed yet.

Aerial photographs, from 1945 and onwards, show that despite partial restoration of water flow after the major drainage effort in 1960, later constructions such as the inner dike and perimeter canals remain in the area. This has limited fresh water inputs and exchanges. Climatic factors caused an annual fresh water deficit of approximately 200 mm. Over time, this deficit has predominately been met by sea water inputs, which has led to increased salinity in the wetland. The installation of the two sluice gates in 1998–1999 likely reduced the rate of the salinity increase. However, we suggest that these fresh water inputs were not enough to restore the less saline, pre-intervention conditions. Compared to climate driven, long-term changes in salinity that have been recorded in the GLw during the Holocene [37], the recent human alterations have resulted in short-term but more dramatic changes.

Freshwater availability in a wetland affects the water salinity, which in turn is critical for the habitats of many species: including fish, amphibians and water-birds [38]. Before 1960, part of the wetland (marshes) was covered with *Typha spp.*, in line with our suggestion that the salinity in the wetland was originally low but increased after the interventions in the 1960s. Even though *Typha* species are invasive, dominating over native aquatic species under moderate salinity, they can germinate only after being returned to non-saline conditions [39], and the most salt-tolerant species (*Typha domingensis*) can only tolerate salinities up to 15 ppt [40,41]. In addition, rice, which according to the KIs and historical maps, was also cultivated in the area prior to 1960, is a salt-sensitive crop that becomes unproductive when soil salinity reaches 7 ppt [42].

We thus suggest that, over the years, the combined effect of increasing salinity (after 1976) and limitation in water circulation (after 1984–1985) has led to extensive reed and cattail mortality and expansion of halophytic species. As a result, the fresh and brackish water marshes have either been covered by open water or gradually been replaced by halophytic vegetation.

Salinity increases and subsequent ecosystem changes could be reversed. For example, a transition from saline to more brackish conditions was observed in another Mediterranean wetland based on the expansion of fresh and brackish water wetland species (*Sarcocornia fruticosa, Phragmites australis* and *Juncus maritimus*) [31]. However, for such a reversal to occur in the GLw, the original hydrologic flows would need to be re-established.

In addition, some areas at the margin of the wetland have been converted to agriculture land. Similar to the GLw case, increasing human demand for land and for fresh water has driven the disappearance of 50% of Mediterranean wetlands during the 20th century [32]. The transformation

of wetlands to agricultural land has been observed in other areas around the Mediterranean basin during the period 1975–2005 [33], and similar impacts of human interventions on wetlands have been observed worldwide. Alterations of hydrologic connectivity and consequent establishment of hypersaline conditions have caused widespread mangrove mortality in tropical wetlands [34].

The loss and transformation of wetlands affects animal populations as well. These changes are among the reasons for which declining water-bird populations in the last century have been attributed [43–45]. Compared to salt marshes, reeds and cattail habitats provide food for water birds, protection from their predators and a sheltered area for nesting [7]. The reduction and transformation of fresh water wetlands affect bird species abundance, diversity and distribution. The KIs mentioned that ducks, geese, swans and other water birds were much more common in the past than today. Indeed, recent studies indicate that the number of water birds had decreased during the period 1986–2015. A notable except to this trend were flamingo populations, which was the only species with a clear positive trend [46]. As in the GLw, the populations of flamingos have also increased in other Mediterranean areas during the last decades [47]. As flamingos feed on and breed next to saline lakes and prefer saline waters [48], their increase in the GLw could be linked to the increased salinity.

Reduced fresh water inputs over time may also have affected mammals, amphibians and water turtles [7]. During fieldwork, we observed turtles and frogs in the two perimeter canals but none in the southern wetland area, which was once covered with reeds and cattail. Additionally, the KIs mentioned that otters (*Lutra lutra*) were frequently seen in the area before 1960, but not thereafter. The presence of otter was recently inferred by the authors from excreta examination, and this observation was verified by local fishermen who have occasionally seen this mammal. Therefore, the species is most likely present, but in low numbers. Increased salinity values in the lagoon were also correlated to fishing yields [49] and benthic community distribution and status [10,15]. Dystrophic crises, documented in the lagoon during 1995–1996, have increased in size and duration during 1998–1999, after the opening of the freshwater inflow culverts [9], possibly due to decreased fresh water quality and quantity.

The changes observed in the lagoon and wetland areas were partly due to intensified land and resource uses. Monocultures of olive trees and coastal tourism were identified as the two major current economic activities. In the wider area, most of the olive farms are not organic [50] and the use of agrochemicals possibly leads to increased inputs of nutrients and pesticides to the water bodies that feed the wetland. Furthermore, compared to the past, the higher water demands for agriculture and tourism during peak season increases the pressure on groundwater. Water used by the olive oil industry could further negatively affect the water quantity in rivers and aquifers in the area [51]. At present, the KIs refer to the Xerolagados stream as "the black river" because it is regularly polluted with olive mill waste waters. However, further investigations are needed to characterize the extent of contamination stemming from this waste water. In contrast, the Tyflomitis area was mentioned as "the lung of the wetland". As evident in the hydrogeological map (Figure S3), the Tyflomitis aquifer (which provides water to the homonymous artesian springs) is mainly located under the Eleofito hills. This area is predominantly covered by olive tree plantations and scattered houses, whose water demands are met by the Tyflomitis groundwater, potentially decreasing water flows and causing contamination (Figure 1).

Lower groundwater flow may cause saltwater intrusions. In previous studies, local residents have mentioned that salinity in the old wells near GLw was almost too high to use water for drinking [51]. In the current study, the KIs mentioned that in the last 15–20 years' "groundwater is becoming more brackish". Older studies [22,28] report reduced water quality and increased values of conductivity and $Cl^-$ in the groundwater, both indicating possible seawater intrusion into the freshwater aquifer due to increased water abstraction. Apart from wells used by the Pylos-Nestor municipality for water supply (three with total pumping capacity up to $1.12 \times 10^6$ m$^3$/year), we observed that there were many more informal wells in the area. Our analysis shows that climate change was relatively minor in the area (1956–2011). However, with the drier conditions expected in the future [52], increased groundwater demand could lead to increased risk for seawater intrusion into Tyflomitis aquifer. Seawater intrusion

is already threatening other Mediterranean coastal aquifers [53,54], and future hydrologic studies in the area should provide a better understanding on the status of water resources and uses.

Furthermore, water management associated with the expansion of tourism coupled with the effect of climate change, could lead to the reduction of groundwater storage and streamflow [35], resulting in environmental, social and economic impacts in the area [55]. Tourism is expanding together with infrastructure development (hotels, roads and airports), providing opportunities for diversified livelihoods, but also increasing pressures on the environment and historical sites. Inside the Glw-Natura2000 area, not only the size of the Divari sand barrier size has been reduced by agriculture expansion (Table 1), but recently, uncontrolled tourism activities pose additional threats to the sand dune ecology. The wider coastal area is the basic nesting habitat for the only European population of the African Chameleon (*Chamaeleo africanus*), a critically endangered (CR) species [56]. The nesting habitat of the species is limited to the sand dunes of Divari, and its survival depends on the management and protection of these coastal dunes [57]. The chameleons are particularly active during the breeding and nesting period (July–September) [58], which coincides with the peak touristic season, and previous studies report several chameleon deaths due to motor vehicles [59]. Their nests are threatened by vehicle activity on the sand dunes all year around, since the incubation lasts for 11 months and their eggs are buried in the sand for the entire period [59].

After being a subject of human alterations from 1960 onwards, we suggest that the GLw has passed the tipping point of being a brackish wetland and has been transformed into a seasonally saline wetland, with profound implications in the area's habitats and species populations. Even so, the area is still considered to be an important area in Europe for many flora and fauna species [7,17,46,56,57], and is today included in the Natura 2000 network of protected areas. Nonetheless, the implementation of this management framework (Natura 2000) has faced [17], and is still facing, major difficulties, creating conflicts among the different stakeholders. For example, conflicting water needs between farmers and fishermen using the lagoon were reported in the SES as a limiting factor for the management of flow regulation via the gate systems [17]. Conflicts regarding land uses inside the GLw-Natura2000 have also been intense in the past [17].

Sustainable Development Goal (SDG) 17, "Partnerships for achieving the goals" [60], is a cornerstone for building a sustainable society. Dynamic ecosystems, such as coastal wetlands, need adaptive co-management strategies, where the needs of different stakeholders can be accommodated in a multifunctional landscape to create win-win situations and reduce conflicts, in contrast to past management efforts. However, even at present, this is not an easy task. Under the current economic situation in Greece, primary production (such as agriculture and fishing) and tourism are among the pillars of economic development. In the area, agriculture is a traditional and major economic activity, while tourism offers opportunities for additional income. Currently local stakeholders do not perceive the protection/conservation and management of the Natura 2000 area as a basis for sustainable economic development, but rather as a constraining factor (based on authors discussions with local residents from 2010 onwards). Wetlands provide a variety of important ecosystem services (ES) to the whole society [3,61] and contribute to many of the 17 SDGs, either directly or indirectly. Therefore, the conservation and sustainable use of wetlands represent a cost-effective investment for governments to meet these goals [62].

In January 2019, the Management Body of Protected Areas of South Peloponnese and Kythira Island (MBPASPK), was established to fill the institutional gap that existed [63]. The MBPASPK has the mandate to first evaluate and update the guidelines of the old SES (i.e., for water uses, conservation of species and habitats, and human activities such as fishing, agriculture and tourism), and subsequently to apply them and manage the area. For example, the area has the potential to increase eco-tourism activities, such as bird-watching [17], which could increase awareness among the stakeholders and add to the local economy by attracting visitors during the less touristic months (November–March). However, increased number of visitors could affect the environment even more, if not organized and managed correctly. A detailed and updated bird survey could provide the basis

for conservation strategies, but also suggestions for activities such as birdwatching, eco-tours and environmental education.

Such management requires not only interdisciplinary research, but also engagement of stakeholders at a broader scale than in the past. The creation of a multi-actor laboratory (MAL) platform, funded by the COASTAL (Collaborative Land-Sea Integration Platform) project, has brought together actors from different sectors (i.e., agriculture, fishing, local industry, tourism and public sectors) in SW Messinia, and under this umbrella, local stakeholders have met for the first time to discuss land–sea interactions, and how these can be improved in the future [64]. ES allow identifying links between nature and people [65], and we suggest that stakeholders' engagement could be fostered by leveraging the value of ES provided by the whole Natura 2000 site (as in References [66–69]), eventually improving the management of the area.

## 5. Conclusions

This study shows how major human interventions and climatic factors during the last century have altered the characteristics of the Gialova Lagoon wetland by transforming its hydrologic and ecological functions. Lacking continuous water quality data and detailed cartographic references, we synthesized available evidence (land cover/land uses changes, climatic data) with a conceptual hydrologic model and knowledge from local elderly. By using a conceptual hydrologic and salt mass model linked to climatic factors, field measurements and aerial photographs analysis, we show that human interventions have reduced freshwater inputs and water circulation in the wetland, resulting in increased salinity. Based on our land cover/land use analysis, we suggest that these hydrologic changes have altered wetland habitats, which have transitioned from being freshwater species-dominated to saline species-dominated. This has likely affected the abundance and distribution of birds and other species. Moreover, the natural environment of the wetland has been under stress from human interventions within the wetland, and also from activities such as agriculture and tourism in surrounding lands. By considering these activities and their consequences over a 70-year period, our results provide a historical perspective on the socio-hydrological and socio-ecological dynamics that have caused dramatic shifts in the function of the Gialova Lagoon wetland. We suggest that any future management plan should be based on a more holistic approach considering climate scenarios, as well as land uses, local economic activities and relationships with natural resources, ultimately generating case-specific and sustainable solutions.

**Supplementary Materials:** The following are available online at: http://www.mdpi.com/2073-4441/11/2/350/s1. Figure S1: 1952 historical map; Figure S2: 1945 aerial photograph on which past uses and vegetation were drawn by the KIs; Figure S3: Hydrogeological map; Figure S4: The history of agriculture in Messinia for the period 1964–2006; Figure S5: Number of foreign and Greek tourist arrivals; Table S1: Area coverage (hectares) of different classes in GLw-Natura2000 and the Tyflomitis area, in reference (prior to 1960) and present (2010) periods; Table S2: Area coverage (hectares) of different classes in the wider area in reference (prior to 1960) and present (2010) periods.

**Author Contributions:** Conceptualization, G.M., H.B., D.B. and S.M.; Methodology, G.M., D.B. and S.M.; Software, E.M. and S.M.; Validation, G.M., H.B. and S.M.; Formal analysis, G.M., E.M. and S.M.; Investigation, G.M. and D.B.; Data curation, G.M., E.M. and S.M.; Writing—original draft preparation, G.M.; Writing—review and editing, G.M., H.B. and S.M.; Visualization, G.M., E.M. and S.M.; Supervision, H.B. and S.M.; Project administration, G.M., H.B. and S.M.

**Funding:** The authors acknowledge support from the Navarino Environmental Observatory (NEO), a partnership between Stockholm University, the Biomedical Research Foundation of the Academy of Athens (BRFAA) and TEMES S.A. G.M. and H.B. acknowledge COASTAL (H2020-RUR-02-2017, No. 773901). S.M. was partly supported by the Swedish Research Councils (Vetenskapsrådet, Formas) and Sida joint project VR 2016-06313.

**Acknowledgments:** The authors would like to thank the participants to interviews for their time and contribution and the National Observatory of Athens for providing climatic data. Estela Ruiz de Azua Sudupe, Lidia Roncero Crespo, Julie Lauterbach and Victor-Emmanuel Le Cunff provided invaluable help in the field (2010–2011). We also thank John Livsey and Martha Papathanassiou for proofreading the text. Valuable comments were provided by two anonymous reviewers.

**Conflicts of Interest:** The authors declare no conflict of interest.

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
