# Peer review of "Anthropogenic Changes in a Mediterranean Coastal Wetland during the Last Century—The Case of Gialova Lagoon, Messinia, Greece"

_water, doi:10.3390/w11020350_

Round 1

Reviewer 1 Report

In general terms, the research is very well developed. The addressed topic is focal and of great interest for Water readers. It is in line with the objectives of the Special Issue "Wetlands and Their Roles in the Ecohydrological Cycle under Global Climate Change" and with the section "Water Quality and Ecosystems".

The manuscript describes the dynamics of a Mediterranean wetland and its surroundings, altered by changes in hydrological, climatic conditions and land uses that have occurred over almost a century. These biophysical and anthropogenic changes have important consequences on habitats and species, some of which are endangered. It is not an isolated case study, but is a good example of what happens in many other Mediterranean wetlands, belonging to the Natura 2000 Network. The holistic approach, which includes the biophysical and socio-economic aspects, is very relevant. The semi-experimental method used, type BACI (Before-After-Control-Impact), is very appropriate although the authors do not mention it explicitly.

The structure of the manuscript is correct and standard. The bibliographical references are abundant and adequate. The "discussion" section is very complete, comparing and discussing the results obtained with other cases of study in the Mediterranean area and other continents.

Nevertheless, a minor review is essential in order to improve some formal and content aspects of the manuscript. Below, a series of suggestions are proposed as a guide for the review:

ü  In section 2.2.1. It would be convenient for the authors to explain, briefly, what they mean with the expression “interpreted with conceptual schematics models” (lines 95-96).

ü  In line 167, authors mention “described in section 2.3.2.”  However, this section seems to be inexistent. Do the authors refer to section 2.2.1?

ü  In figure 5, I suggest to unify the toponym of Giannozagas vs Giannouzagas. Which one is correct? Also, I recommend relocating this toponym on the aerial photographs in figure 5a to improve its readability. At the bottom of this figure I recommend replacing the reference to "period 1999-2009" with "period 2000-2009".

ü  To facilitate reading, in addition to the colour key in the y-axis of figure 7, I recommend adding the initials of the climatic variables represented at the right of their corresponding curves.

ü  In general, table 1 is correct to express the trends of each land use-land cover. However, with a little more effort, perhaps the authors could address the superposition of the two maps (reference period vs. present period) using the cross-tabulation technique. In this way, they would provide the reader with more information: for example, the nature of the change (from class of origin to class of destination) and the usual statistics in other similar articles (persistence, gains, losses, total change, net change and annual rate of change for each class).

ü  Consider the convenience of including some reference to the book chapter: Tiniakos, L., Tiniakou, A. 1997. Geomorphological processes and environmental considerations in the Navarino Bay area, SW Peloponnisos, Greece, in M. Koukis, Tsiambaos & Stoumaras (Eds.) Engineering Geology and the Environment, Rotterdam, Balkema, 397-401.

Author Response

Thank you for your support and the constructive comments, which we address point-by-point in the following. The point-by-point answers are given based on the clean revised manuscript.

Giorgos Maneas, on behalf of all authors.

Point 1: In general terms, the research is very well developed. The addressed topic is focal and of great interest for Water readers. It is in line with the objectives of the Special Issue "Wetlands and Their Roles in the Ecohydrological Cycle under Global Climate Change" and with the section "Water Quality and Ecosystems".

The manuscript describes the dynamics of a Mediterranean wetland and its surroundings, altered by changes in hydrological, climatic conditions and land uses that have occurred over almost a century. These biophysical and anthropogenic changes have important consequences on habitats and species, some of which are endangered. It is not an isolated case study, but is a good example of what happens in many other Mediterranean wetlands, belonging to the Natura 2000 Network. The holistic approach, which includes the biophysical and socio-economic aspects, is very relevant. The semi-experimental method used, type BACI (Before-After-Control-Impact), is very appropriate although the authors do not mention it explicitly.

The structure of the manuscript is correct and standard. The bibliographical references are abundant and adequate. The "discussion" section is very complete, comparing and discussing the results obtained with other cases of study in the Mediterranean area and other continents.

Response 1: Indeed, our methodology is similar to BACI methodology but follows a more qualitative approach, due to lack of consistent data throughout the study period. Approaching these issues with a quantitative BACI methodology is an excellent suggestion for future work. Our mixed-methods approach could be useful for similar studies in areas with limited available data over time. We also hope that the links between human activities and wetland dynamics proposed can stimulate discussion and further studies in this area. To improve context, this case study is now discussed in the context of other case studies in the Mediterranean region. Further below, we provide answers on the proposed changes suggested.

Point 2: In section 2.2.1. It would be convenient for the authors to explain, briefly, what they mean with the expression “interpreted with conceptual schematics models” (lines 95-96).

Response 2: We added an explanation on what we meant (line 114-116)

Point 3: In line 167, authors mention “described in section 2.3.2.”  However, this section seems to be inexistent. Do the authors refer to section 2.2.1?

Response 3: We have changed to 2.2.1 (line 187).

Points 4-5:

- In figure 5, I suggest to unify the toponym of Giannozagas vs Giannouzagas. Which one is correct? Also, I recommend relocating this toponym on the aerial photographs in figure 5a to improve its readability. At the bottom of this figure I recommend replacing the reference to "period 1999-2009" with "period 2000-2009".

- To facilitate reading, in addition to the colour key in the y-axis of figure 7, I recommend adding the initials of the climatic variables represented at the right of their corresponding curves.

Response 4-5: We have revised all relevant figures with the corrected toponym (Figures 3-5) and also added some extra information in schematics. Also, the initials of the climatic variables in Figure 7 have been added. Furthermore, we have updated Figure 1 to improve readability and cover in more detail the area of our case study, and Figure 8 to correct a minor mistake that we found in the map.

Point 6: In general, table 1 is correct to express the trends of each land use-land cover. However, with a little more effort, perhaps the authors could address the superposition of the two maps (reference period vs. present period) using the cross-tabulation technique. In this way, they would provide the reader with more information: for example, the nature of the change (from class of origin to class of destination) and the usual statistics in other similar articles (persistence, gains, losses, total change, net change and annual rate of change for each class)

Response 6: In the revised version we provide 2 tables providing more information on past to present changes in both GLw-Natura2000 area and the wider are of interest. We hope that in this way we address the suggested improvements.

Point 7: Consider the convenience of including some reference to the book chapter: Tiniakos, L., Tiniakou, A. 1997. Geomorphological processes and environmental considerations in the Navarino Bay area, SW Peloponnisos, Greece, in M. Koukis, Tsiambaos & Stoumaras (Eds.) Engineering Geology and the Environment, Rotterdam, Balkema, 397-401.

Response 7: We tried to get full-access to the proposed book chapter, but it is not available at the Stockholm University library. We also contacted the authors, but so far we did not get any response. We will keep trying to get access and add a reference to the final version.

Additional References

1.        Apostolopoulou, E.; Pantis, J. D. Development Plans versus Conservation: Explanation of Emergent Conflicts and State Political Handling. Environ. Plan. A 2010, 42 (4), 982–1000. https://doi.org/10.1068/a42163.

2.        Álvarez-Rogel, J.; Jiménez-Cárceles, F. J.; Roca, M. J.; Ortiz, R. Changes in Soils and Vegetation in a Mediterranean Coastal Salt Marsh Impacted by Human Activities. Estuar. Coast. Shelf Sci. 2007, 73 (3–4), 510–526. https://doi.org/10.1016/j.ecss.2007.02.018.  

3.        Perennou, C.; Beltrame, C.; Guelmami, A.; Tomas Vives, P.; Caessteker, P. Existing Areas and Past Changes of Wetland Extent in the Mediterranean Region: An Overview. Ecol. Mediterr. 2012, 38 (2), 53–66.

4.        Mediterranean Wetlands Observatory, 2014. Land cover - Spatial dynamics in Mediterranean coastal wetlands from 1975 to 2005. Thematic collection, issue #2. Tour du Valat, France. 48 pages. Translated by Charles La Via and Adam Clark. ISBN: 2-910368-60-2.

5.        Jaramillo, F.; Licero, L.; Åhlen, I.; Manzoni, S.; Rodríguez-Rodríguez, J. A.; Guittard, A.; Hylin, A.; Bolaños, J.; Jawitz, J.; Wdowinski, S.; et al. Effects of Hydroclimatic Change and Rehabilitation Activities on Salinity and Mangroves in the Ciénaga Grande de Santa Marta, Colombia. Wetlands 2018, 38 (4), 755–767. https://doi.org/10.1007/s13157-018-1024-7.

6.        Handrinos, G.; Kazantzidis, S.; Alivizatos, C.; Akriotis, T.; Portoliou, D. International Waterbird Census in Greece (1968-2006); Hellenic Ornithological Society - Hellenic Bird Ringing Centre: Athens, 2015.

7.        Transforming our world: the 2030 Agenda for Sustainable Development. Available online: https://sustainabledevelopment.un.org/post2015/transformingourworld (accessed in 23 January 2019).

8.        Reid, W. V; Mooney, H. A.; Cropper, A.; Capistrano, D.; Carpenter, S. R.; Chopra, K.; Dasgupta, P.; Dietz, T. Ecosystems  and Human Well-Being; 2005. https://doi.org/10.1196/annals.1439.003.

9.        Wetlands and the SDGs. 2018. Scaling up wetland conservation, wise use and restoration to achieve the Sustainable Development Goals. Ramsar Convention On Wetlands. Available report on line: https://www.ramsar.org/document/wetlands-and-the-sdgs  (downloaded in 23 January 2019).

10.      Berg, H.; Ekman Söderholm, A.; Söderström, A.-S.; Tam, N. T. Recognizing Wetland Ecosystem Services for Sustainable Rice Farming in the Mekong Delta, Vietnam. Sustain. Sci. 2017, 12 (1), 137–154. https://doi.org/10.1007/s11625-016-0409-x.

11.      McInnes, R. J.; Everard, M. Rapid Assessment of Wetland Ecosystem Services (RAWES): An Example from Colombo, Sri Lanka. Ecosyst. Serv. 2017, 25, 89–105. https://doi.org/10.1016/j.ecoser.2017.03.024.

12.      Sharma, B.; Rasul, G.; Chettri, N. The Economic Value of Wetland Ecosystem Services: Evidence from the Koshi Tappu Wildlife Reserve, Nepal. Ecosyst. Serv. 2015, 12, 84–93. https://doi.org/10.1016/j.ecoser.2015.02.007.

13.      Janse, J. H.; van Dam, A. A.; Hes, E. M. A.; de Klein, J. J. M.; Finlayson, C. M.; Janssen, A. B. G.; van Wijk, D.; Mooij, W. M.; Verhoeven, J. T. A. Towards a Global Model for Wetlands Ecosystem Services. Curr. Opin. Environ. Sustain. 2019, 36 (September 2018), 11–19. https://doi.org/10.1016/j.cosust.2018.09.002.

14.      Díaz, S.; Demissew, S.; Carabias, J.; Joly, C.; Lonsdale, M.; Ash, N.; Larigauderie, A.; Adhikari, J. R.; Arico, S.; Báldi, A.; et al. The IPBES Conceptual Framework - Connecting Nature and People. Curr. Opin. Environ. Sustain. 2015, 14 (June), 1–16. https://doi.org/10.1016/j.cosust.2014.11.002.

Reviewer 2 Report

An interesting study topic. The paper analyses well the specific situation in the study area but presented the research in a very limited perspective without providing a contextualization in light of international, up-to date research on impacts on wetland areas (e.g. climate change related impacts, management of Natura 2000 wetland areas etc).

In particular the methodological approach should be explained and discussed with regard to recent literature e.g. Janssen et al. 2019. Please add a sound background section and discuss these international perspectives in the current section two and four as well. 

It would be helpful to read about the endangered species not only in the discussion section but already in the first section(s). Explaining a bit more about the Natura 2000 area itself (and the management plan/targets) could increase the understanding of the importance of certain aspects of the whole wetland area respectively the consequences of their degradation right at the beginning (e.g. of the marshland area).

Impacts of climate change are only discussed at the very final part of the paper, they might merit an earlier consideration however. 

Despite a very detailed view at the constructional and structural changes, the detailed impact of  human influences, in particularly the recent ones - tourism and intensified agricultural production with irrigation - seem not to be entirely/appropriately captured in the model. 

It would be beneficiary to provide hints about balancing the major interests with protection targets in a more concrete way instead of just acknowledging the potential for conflicts without any concrete suggestions for the area. Finally Natura 2000 management plans are mentioned -  the reader would appreciate information how water extraction etc. are treated/ considered in the management targets and negiotiation with actors. 

Author Response

Thanks for your positive comments and constructive critiques, which we address point-by-point in the following. The point-by-point answers are given based on the clean revised manuscript.

Giorgos Maneas, on behalf of all authors.

Point 1: The paper analyses well the specific situation in the study area but presented the research in a very limited perspective without providing a contextualization in light of international, up-to date research on impacts on wetland areas (e.g. climate change related impacts, management of Natura 2000 wetland areas etc).

Response 1: Thank you for this comment. We have added text and references in the discussion to expand our perspective and link more to international, up-to date research on impacts on wetland areas. Several references to case studies in the Mediterranean and beyond have also been added. Main changes in response to this comment: lines 488-492; 497-503; 591-597.

Point 2: In particular the methodological approach should be explained and discussed with regard to recent literature e.g. Janssen et al. 2019. Please add a sound background section and discuss these international perspectives in the current section two and four as well.

Response 2: We have added text and references to link our approach to recent literature as suggested (lines 591-597). However, we found it difficult to compare our methodology to the suggested literature since at the moment there is lack of consistent data for our case study, which limits our efforts in applying more quantitative approaches and models. Nonetheless, our approach provides background information linking human activities to wetland dynamics in the area for the first time. Most important, all pieces of evidence we collected are consistent, offering an integrated (a new in this area) perspective.

Point 3: It would be helpful to read about the endangered species not only in the discussion section but already in the first section(s). Explaining a bit more about the Natura 2000 area itself (and the management plan/targets) could increase the understanding of the importance of certain aspects of the whole wetland area respectively the consequences of their degradation right at the beginning (e.g. of the marshland area).

Response 3: Thank you for this comment. We have added a new paragraph on the history and management challenges of the Natura 2000 area in the Introduction section (lines 47-60) and improved other parts (lines 61-69) to provide a better explanation to the reader, as well as present the current situation.

Points 4: Impacts of climate change are only discussed at the very final part of the paper, they might merit an earlier consideration however.

Response 4: We showed that climatic changes were relatively minor in the area, with precipitation trends that were not statistically significant (Section 3.1.2). Because of this limited climate change influences, we focused this work on historical anthropogenic impacts. A complete account of predicted climatic changes is outside our scope here, and would be highly speculative given the high uncertainties in particular regarding changing in wetland management in response to future climatic changes. As a result, climate change aspects are brought up and discussed toward the end of the paper as part of the outlook for the future (lines 618-621). Links to climate change also discussed in lines 468-471; 475-476; 553-556; 617-620.

Point 5: Despite a very detailed view at the constructional and structural changes, the detailed impact of human influences, in particularly the recent ones - tourism and intensified agricultural production with irrigation - seem not to be entirely/appropriately captured in the model

Response 5: The reviewer is correct in saying that our conceptual model is not able to capture the details of human interventions and their consequences. We did not develop a process model that could quantify hydrologic changes as a result of human interventions because we lack the basic information to do so. Our historical account is detailed, but in part qualitative, allowing to frame human interventions in the context of minimal water and salt mass balance equations. This minimal approach gives some insights on the expected dynamics of the lagoon system, based on rough estimates of the historical changes in hydrologic fluxes. At this stage, we cannot offer a more quantitative and detailed approach. However, we emphasized this limitation in the Conclusions: “Lacking continuous water quality data and detailed cartographic references, we synthesized available evidence (land cover/land uses changes; climatic data) with a conceptual hydrologic model that also leverages local knowledge from local elderlies.”.

Point 6: It would be beneficiary to provide hints about balancing the major interests with protection targets in a more concrete way instead of just acknowledging the potential for conflicts without any concrete suggestions for the area.

Response 6: Indeed, discussing concrete ways to address conflicts is a good idea and we have added some suggestions to this regard in the final parts of the Discussion. However, the purpose of this paper is not to provide suggestions to improve the management, but make the point that management can be improved by more active participation of stakeholders, compared to past practices. A concrete action plan can be developed based on the background historical information we gathered in this contribution and will be the topic of future work.

Point 7: Finally Natura 2000 management plans are mentioned -  the reader would appreciate information how water extraction etc. are treated/ considered in the management targets and negiotiation with actors.

Response 7: Thank you for this comment. We have added text in the discussion (lines 575-580) and also background information in the Introduction section (lines 47-60).

 Additional References

1.        Apostolopoulou, E.; Pantis, J. D. Development Plans versus Conservation: Explanation of Emergent Conflicts and State Political Handling. Environ. Plan. A 2010, 42 (4), 982–1000. https://doi.org/10.1068/a42163.  (19)

2.        Álvarez-Rogel, J.; Jiménez-Cárceles, F. J.; Roca, M. J.; Ortiz, R. Changes in Soils and Vegetation in a Mediterranean Coastal Salt Marsh Impacted by Human Activities. Estuar. Coast. Shelf Sci. 2007, 73 (3–4), 510–526. https://doi.org/10.1016/j.ecss.2007.02.018.  

3.        Perennou, C.; Beltrame, C.; Guelmami, A.; Tomas Vives, P.; Caessteker, P. Existing Areas and Past Changes of Wetland Extent in the Mediterranean Region: An Overview. Ecol. Mediterr. 2012, 38 (2), 53–66.

4.        Mediterranean Wetlands Observatory, 2014. Land cover - Spatial dynamics in Mediterranean coastal wetlands from 1975 to 2005. Thematic collection, issue #2. Tour du Valat, France. 48 pages. Translated by Charles La Via and Adam Clark. ISBN: 2-910368-60-2.

5.        Jaramillo, F.; Licero, L.; Åhlen, I.; Manzoni, S.; Rodríguez-Rodríguez, J. A.; Guittard, A.; Hylin, A.; Bolaños, J.; Jawitz, J.; Wdowinski, S.; et al. Effects of Hydroclimatic Change and Rehabilitation Activities on Salinity and Mangroves in the Ciénaga Grande de Santa Marta, Colombia. Wetlands 2018, 38 (4), 755–767. https://doi.org/10.1007/s13157-018-1024-7.

6.        Handrinos, G.; Kazantzidis, S.; Alivizatos, C.; Akriotis, T.; Portoliou, D. International Waterbird Census in Greece (1968-2006); Hellenic Ornithological Society - Hellenic Bird Ringing Centre: Athens, 2015.

7.        Transforming our world: the 2030 Agenda for Sustainable Development. Available online: https://sustainabledevelopment.un.org/post2015/transformingourworld (accessed in 23 January 2019).

8.        Reid, W. V; Mooney, H. A.; Cropper, A.; Capistrano, D.; Carpenter, S. R.; Chopra, K.; Dasgupta, P.; Dietz, T. Ecosystems  and Human Well-Being; 2005. https://doi.org/10.1196/annals.1439.003.

9.        Wetlands and the SDGs. 2018. Scaling up wetland conservation, wise use and restoration to achieve the Sustainable Development Goals. Ramsar Convention On Wetlands. Available report on line: https://www.ramsar.org/document/wetlands-and-the-sdgs  (downloaded in 23 January 2019).

10.      Berg, H.; Ekman Söderholm, A.; Söderström, A.-S.; Tam, N. T. Recognizing Wetland Ecosystem Services for Sustainable Rice Farming in the Mekong Delta, Vietnam. Sustain. Sci. 2017, 12 (1), 137–154. https://doi.org/10.1007/s11625-016-0409-x.

11.      McInnes, R. J.; Everard, M. Rapid Assessment of Wetland Ecosystem Services (RAWES): An Example from Colombo, Sri Lanka. Ecosyst. Serv. 2017, 25, 89–105. https://doi.org/10.1016/j.ecoser.2017.03.024.

12.      Sharma, B.; Rasul, G.; Chettri, N. The Economic Value of Wetland Ecosystem Services: Evidence from the Koshi Tappu Wildlife Reserve, Nepal. Ecosyst. Serv. 2015, 12, 84–93. https://doi.org/10.1016/j.ecoser.2015.02.007.

13.      Janse, J. H.; van Dam, A. A.; Hes, E. M. A.; de Klein, J. J. M.; Finlayson, C. M.; Janssen, A. B. G.; van Wijk, D.; Mooij, W. M.; Verhoeven, J. T. A. Towards a Global Model for Wetlands Ecosystem Services. Curr. Opin. Environ. Sustain. 2019, 36 (September 2018), 11–19. https://doi.org/10.1016/j.cosust.2018.09.002.

14.      Díaz, S.; Demissew, S.; Carabias, J.; Joly, C.; Lonsdale, M.; Ash, N.; Larigauderie, A.; Adhikari, J. R.; Arico, S.; Báldi, A.; et al. The IPBES Conceptual Framework - Connecting Nature and People. Curr. Opin. Environ. Sustain. 2015, 14 (June), 1–16. https://doi.org/10.1016/j.cosust.2014.11.002.

Round 2

Reviewer 1 Report

Dear authors,

The new version of the manuscript responds appropriately to the indications and suggestions of my review. However, before its final publication, I still propose that you make some minor modifications on tables 1 and 2.

I suggest reordering table 1 so that the classes and subclasses are listed in the same order, both in rows and in columns. In this way, the surfaces that do not change (persistence) will be displayed on the diagonal (it is convenient to highlight the cells in gray and the numbers in bold, for example).

Meanwhile, the changes between different uses (gains and losses) will be shown in the cells outside the diagonal. In addition, I recommend that the surfaces are not shown in hectares, but are calculated in% with respect to the total area of the territory studied. I attach an pdf file as a guide.

On the other hand, to facilitate the interpretation of the table it is convenient to mention each class in a similar way both in the reference period and its corresponding one in the current period (Marshes vs Salt meadows). Also check the "Size (ha)" column. The total sum does not equal the sum of all the cells in the same row.

Likewise, in table 2 I suggest indicating the surfaces of the matrix in% with respect to the total surface.

Author Response

Thank you for your support, the constructive comments, and your suggestions which we have improved our manuscript. Please find below our answers to your suggestions.

Giorgos Maneas, on behalf of all authors.

Point 1: The new version of the manuscript responds appropriately to the indications and suggestions of my review. However, before its final publication, I still propose that you make some minor modifications on tables 1 and 2.

I suggest reordering table 1 so that the classes and subclasses are listed in the same order, both in rows and in columns. In this way, the surfaces that do not change (persistence) will be displayed on the diagonal (it is convenient to highlight the cells in gray and the numbers in bold, for example).

Meanwhile, the changes between different uses (gains and losses) will be shown in the cells outside the diagonal. In addition, I recommend that the surfaces are not shown in hectares, but are calculated in% with respect to the total area of the territory studied. I attach an pdf file as a guide.

On the other hand, to facilitate the interpretation of the table it is convenient to mention each class in a similar way both in the reference period and its corresponding one in the current period (Marshes vs Salt meadows). Also check the "Size (ha)" column. The total sum does not equal the sum of all the cells in the same row.

Likewise, in table 2 I suggest indicating the surfaces of the matrix in% with respect to the total surface.

Response 1: Thank you for this suggestion. In the revised manuscript (version 2.1) we have changed the tables following your suggestions and now we provide 2 tables in which the classes are ordered consistently in rows and columns. Persistent surfaces are reported on the diagonal, and surfaces whose class has changed in the other positions in the table. Moreover, we provide the area coverage in hectares in the supplementary materials (Tables S1 and S2).

Other changes in the revised manuscript:

In order to further improve our manuscript, we have conducted a proof-reading to improve language, flow and clarity. To that end:

-        The revised manuscript contains several small corrections in language (the manuscript was proofread by a native speaker).

-        We have removed one paragraph between eq. 4 and 5 in section 3.1.3. It described hydrologic changes in the period of transformation, but did not add much to previous statements, and was not useful for the comparison between past and present. We have moved some information from this paragraph in the earlier sections. A second paragraph between eq. 4 and 5, has been moved after eq. 5.

-        We have updated Figures 1 and 8

-        We have slightly re-arranged the text in discussion to improve flow and clarity, and added at the end of discussion new text and references about the area.

-        we have taken out the appendix and now we provide all the additional figures and tables as supplementary materials, which will be available online.

Additional References

1.   Hellenic Government Gazette, YODD 790/31.12.2018. Establishment of the Management Body of Protected Areas of South Peloponnese and Kythira island. (available in Greek as: ΦΕΚ ΥΟΔΔ 790/31.12.2018).

2.   COASTAL - Collaborative Land-Sea Integration Platform. European Union's H2020 research and innovation programme under grant agreement No 773782. Available on-line: https://h2020-coastal.eu/  (accessed in 03/02/2019).

Reviewer 2 Report

Due to the additional information and discussion of implications including the ES approach the paper has been significantly improved.

Author Response

Point 1: Due to the additional information and discussion of implications including the ES approach the paper has been significantly improved.

Thank you for your support and the constructive comments, which we agree that have improved the paper significantly.

In the revised manuscript (version 2.1), minor changes, in part suggested by reviewer 1, were made. We have changed the tables following suggestions from reviewer 1, and now we provide 2 tables in which the classes are ordered consistently in rows and columns. Persistent surfaces are reported on the diagonal, and surfaces whose class has changed in the other positions in the table. Moreover, we provide the area coverage in hectares in the supplementary materials (Tables S1 and S2).

Other changes in the revised manuscript:

In order to further improve our manuscript, we have conducted a proof-reading to improve language, flow and clarity. To that end:

-        The revised manuscript contains several small corrections in language (the manuscript was proofread by a native speaker).

-        We have removed one paragraph between eq. 4 and 5 in section 3.1.3. It described hydrologic changes in the period of transformation, but did not add much to previous statements, and were not useful for the comparison between past and present. We have moved some information from this paragraph in the earlier sections. The second paragraph between eq. 4 and 5, has been moved after eq. 5

-        We have updated Figures 1 and 8

-        We have slightly re-arranged the text in discussion to improve flow and clarity, and added at the end of discussion new text and references about the area.

-        We have taken out the appendix and added all additional figures and tables as supplementary materials which will be available online.

Giorgos Maneas, on behalf of all authors.

Additional References

1.   Hellenic Government Gazette, YODD 790/31.12.2018. Establishment of the Management Body of Protected Areas of South Peloponnese and Kythira island. (available in Greek as: ΦΕΚ ΥΟΔΔ 790/31.12.2018).

2.   COASTAL - Collaborative Land-Sea Integration Platform. European Union's H2020 research and innovation programme under grant agreement No 773782. Available on-line: https://h2020-coastal.eu/  (accessed in 03/02/2019).